# Multilayered SDN security with MAC authentication and GAN-based intrusion detection

**Nanavath Kiran Singh Nayak, Budhaditya Bhattacharyya**[ID]*

School of Electronics Engineering, Vellore Institute of Technology, Vellore, Tamil Nadu, India

* budhaditya@vit.ac.in

## Abstract

Computer networks are highly vulnerable to cybersecurity intrusions. Likewise, software-defined networks (SDN), which enable 5G users to broadcast sensitive data, have become a primary target for vulnerability. To protect the network security against attacks, various security protocols, including authorization, the authentication process, and intrusion detection techniques, are essential. However, there are several intrusion detection strategies, but the most prevalent methods show low accuracy and high false positives. To overcome these problems, this research work presents a novel four-Q curve authentication system based on Media Access Control (MAC) addresses for a multilayered SDN intrusion detection system utilizing deep learning techniques to identify and prevent attacks. The Four-Q curve authentication system leverages elliptic curve cryptography, a high-performance algorithm that improves authentication security and computational efficiency. Initially, Four-Q curve authentication is performed, followed by univariate ensemble feature selection to select optimal switches. Then, the data collected through the switches are classified as normal, assault, and suspect packets based on the Dual Discriminator Conditional Generative Adversarial Network (DDcGAN) approach. Further, the optimization of DDcGAN is accomplished using the Sheep Flock Optimization Algorithm (SFOA), whereas suspicious packets are categorized using the Growing Self-Organizing Map (GSOM). The DDcGAN-based intrusion detection system outperforms the state-of-the-art approaches in terms of accuracy, precision, F1 score, sensitivity, false-positive rate, power consumption, and network throughput. It achieved an accuracy of 98.29%, an F1 score of 0.975, and a precision of 95.8%. The system's true positive rate attained 99.04% at 50% malicious nodes, while the false alarm rate was as low as 2.05% under the same conditions. Moreover, the system exhibits 4.5% energy savings when compared to existing approaches.

**Data availability statement:** All relevant data are within the manuscript and its Supporting Information files.

**Funding:** The author(s) received no specific funding for this work.

**Competing interests:** The authors have declared that no competing interests exist.

## 1. Introduction

Software-defined networking (SDN) provides a dynamic and programmable computing framework to enhance the performance and manageability of computer networks [1]. SDN is a transformative approach in modern networking, enabling centralized management of network resources by decoupling the control plane from the data plane. Unlike traditional architectures, where these planes are tightly integrated, SDN introduces programmability, simplifying network scalability and adaptability while supporting emerging technologies like 5G, the Internet of Things (IoT), and edge computing. It differs from traditional network architecture in that control planes can be installed with a unique directory or several directories, depending on the applications being used in the configuration [2]. It utilizes a centralized controller to establish communication with various network devices and efficiently manage network traffic in accordance with predefined regulations. There are three possible types of centralized control: physical, logical, and hierarchically distributed [3]. Network function virtualization (NFV) is used to manage a large-scale environment and provide enhanced services that are allowed by SDN [4]. Multi-layer computing is a viable option to address some of the shortcomings associated with conventional SDN [5]. The addition of cloud infrastructure to this framework makes it more effective at handling large amounts of data [6]. The "built SDN cloud" offers enhanced adaptability, scalability, and management over network resources using software defined networks. It consists of four levels, each with its own specific functions [7].

Each layer's gadgets are different from one another [8]. These modifications are accomplished to support all 5G users [9]. Despite some limitations, the network accommodates many users, including some opponents [10]. This multi-layered system has security issues because each layer consists of unique devices that are vulnerable to attacks from distinct types of intrusions [11]. The security objectives in this situation are unauthorized access, data modification, access control, integrity, and availability [12,13]. Thus, an SDN architecture based on MAC is recommended to detect and prevent unauthorized access [14]. Conventional SDN designs face notable challenges, including vulnerability to single-point failures in centralized controllers, scalability issues in handling increased traffic, and limitations in detecting evolving threats such as Distributed Denial of Service (DDoS) attacks. To address these shortcomings, multilayer SDN architectures have emerged as a robust solution. By integrating layers with distinct functions and leveraging Network Function Virtualization (NFV) alongside cloud platforms, these architectures enhance adaptability, scalability, and large-scale data processing capabilities. Despite these advancements, the heterogeneity of devices across layers introduces unique security vulnerabilities, necessitating advanced protection mechanisms. In MAC-based SDN architectures, controllers manage the flow of data packets based on the MAC addresses of devices connected to the network, making intrusion detection an important security concern [15]. Network resources can be exhausted by improper packet processing. Therefore, by implementing an effective Intrusion Detection System (IDS) in a MAC-based SDN system, adversaries can be substantially reduced [16].

To safeguard the network against potential attackers, an advanced multi-plane security framework for SDN was designed. An SDN-based IDS uses a centralized controller to monitor network traffic, analyse it, and identify potential security threats [17]. Thus, it is essential to analyze packet attributes and user traffic to secure a system against attacks. In this case, deep learning algorithms are needed for MAC address based SDN due to the limitations of traditional intrusion detection methods [18]. Additionally, different techniques were presented to neutralize other types of assaults. This can be accomplished through deep learning algorithms that utilize meta-heuristics for optimization [19]. DDoS attacks target distinct types of networks. These attacks are recognized at the application, network, and transport layers [20]. To address these problems, a MAC-based SDN architecture with a dual discriminator using conditional generative adversarial network-based intrusion detection systems (DDcGAN-based IDS) has been proposed. This method effectively identifies and classifies network intrusions using a GAN-based generative model. The dual-discriminator conditional generative adversarial network approach improves intrusion detection accuracy, lowers the number of false positives, and creates synthesized information that is very similar to real network traffic. Eventually, this approach will significantly enhance network security. The main findings and contributions of this research are outlined as follows:

- This article suggests the utilization of DDcGAN within an SDN framework for intrusion detection, specifically focusing on MAC address-based authentication. The initial step involves utilizing the Four-Q Curve to authenticate 5G users, effectively addressing the issue of "Man in the Middle Attacks" and ensuring that only authenticated users gain access. Once authenticated, the optimal switch is selected using the univariate ensemble feature selection technique.

- The distributed controller uses the DDcGAN technique to analyse incoming packets and classify them based on the flow characteristics extracted from the packets. At the application layer, suspicious packets are distinguished from ordinary packets and classified as either legitimate or malevolent using a self-organizing map. Normal packets continue to be processed, while malicious packets are rejected. This method effectively detects and identifies DDoS attacks.

- The optimization of DDcGAN is achieved through the utilization of SFOA. In this study, the proposed technique is implemented on the Ns-3 network simulator and compared to other methods, including GRU-RNN, Multi-Layer Perceptron (MLP), and RNN-SDR.

The remaining sections of this research paper are organized in the following manner: Section 2 presents a literature review on intrusion detection in an SDN-based system. Section 3 provides a description of the proposed DDcGAN approach. The results of the suggested technique are discussed in Section 4. Ultimately, Section 5 concludes the overall assessment of the proposal.

## 2. Literature survey

### 2.1. Related work

Numerous studies have been proposed in the literature related to IDS in SDN; among these few recent studies are: Chaganti R. et al. [21] proposed a deep learning (DL) method for SDN-enabled IDS in the IoT. This method uses long short-term memory (LSTM) to detect attacks on the network. However, it requires large amounts of labelled data and a long training time. The performance analysis of this approach shows that the DL method performs better than the ML classifier but needs validation in adversarial environments. Guezzaz et al. [22] developed a reliable intrusion detection system (IDS) for networks incorporating decision tree technology. Pre-processing and feature selection phases were carried out on network data to improve data quality and training effectiveness. The decision tree was built using the entropy-based feature selection and machine learning techniques, but it faces overfitting and limited scalability for complex attack scenarios. Le et al. [23] suggested XGBoost for imbalanced multiclass classification-based IDS in 2022. The raw datasets were pre-processed and evaluated by measuring two performance metrics: the learning curve and the confusion matrix. This method achieved 99.9% and 98.7% of the f1-score for both datasets but needs to be tested and applicable to a particular IIoT application.

Wani et al. [24] developed an intrusion detection system (IDS) for the Internet of Things (IoT) using a deep learning (DL) classifier aided by SDN technology. This method uses an LSTM classifier to detect attacks in a network. The main benefit of this approach is its ability to detect any unauthorized network intrusion, particularly within IoT networks. It is advisable to test the effectiveness of this method in real-time situations with simulated attacks. Perez-Diaz JA et al. [25] presented a versatile SDN-based architecture that utilizes machine learning (ML) to recognize and combat DDoS attacks. The researchers used two distinct topologies to mitigate all the identified attacks. Nevertheless, further advancements in deep learning (DL) and machine learning (ML) techniques need to be combined to improve system performance. Tang TA et al. [26] suggested a gated recurrent unit recurrent neural network (GRU-RNN) enabled IDS for SDN. The detection rate of this approach was 89%, and it achieved better results for both anomalous and legitimate traffic traces. Network performance evaluation shows that GRU-RNN does not significantly affect controller performance, but other features are needed to reduce overhead and increase accuracy.

Albahar MA [27] has developed a real-time intrusion detection system (IDS) in SDN using a recurrent neural network (RNN) and a new regularization method. The system has three main components: a flow collector, an anomaly detector, and an anomaly mitigator. The performance of RNN-SDR showed that it has the potential for real-time detection. This method requires fewer features than other techniques, but it slightly affects the performance of the controller. Scalability to larger, complex environments remains uncertain.

Nguyen TG et al. [28] developed an intelligent and collaborative architecture for SDN-based IoT networks, extracting 30,000 data samples from CAIDA, KDD Cup 1999, and UNSW-NB15 datasets. A novel path selection and system resource optimization have been suggested to reduce communication and resource management overhead. There is a need to explore other potential cyberattacks and deep learning algorithms that can handle large amounts of data.

Phan TV and Park M [29] suggested an efficient DDoS attack defence in the SDN-based cloud. An enhanced History based IP Filtering (eHIPF) scheme was proposed to improve attack detection speed. A new mechanism has been introduced that combines the proposed method and hybrid ML model to make a DDoS attack defender for the SDN based cloud environment. Nonetheless, more evaluation criteria are needed to compare this system to other ML techniques. Scalability and broader applicability need validation. Latah M. and Toker L. [30] proposed an efficient, multi-level hybrid IDS for SDN. The accuracy of this method was 84.29%. Initially, the KNN approach was employed, followed by ELM and H-ELM. The time required for testing was reduced by using ML-based classification algorithms. Improvements are needed to achieve a better accuracy, false alarm rate (FAR) and scalability. Toony et al. [31] developed a multi-block framework, which combines machine learning, stateful P4 processing, and an SDN-based multi-controller architecture to manage critical tasks in IoT networks. The framework includes four key modules: Pyramidal Conceptually Decentralized Multi-Controller Structure (PCDMCS) with Decentralized Control Interfaces (DCIs) for real-time threat detection via a Decentralized Warning Conduit (DWC), advanced network monitoring with P4-enabled 24-state tables, and enhanced anomaly detection using 30 new P4-extracted features and the Enhanced Weighted Ensemble Algorithm (EWEA). It has limited accuracy (84.29%) and a high false alarm rate (FAR). It needs further enhancements for reliability and scalability. The complex multi-module structure requires high computational resources for processing.

Khedr et al. [32] introduced a four-module DDoS attack detection and mitigation framework (FMDADM), an SDN-based DDoS attack detection and mitigation framework for IoT networks. It includes four modules within a five-tier architecture: early detection using the ADR principle, a Double-Check Mapping Function (DCMF) for data plane-level detection, an ML-based detection process with seven features, and attack mitigation. Detection is limited to DDoS attacks and may not generalize to other attacks. Girdler and Vassilakis [33] developed an SDN-based Intrusion Detection and Prevention System (IDPS) designed to defend against ARP spoofing and blacklisted MAC address attacks. The system dynamically adjusts SDN parameters to detect malicious network traffic. Custom software was created for attack testing and IDPS

customization, integrated with a tailored library for validating user input. Enhancements to SDN includes improved attack detection, firewall capabilities, intrusion prevention, packet dropping, and reduced timeout durations. However, the system is limited to ARP spoofing and blacklisted MAC address attacks, requiring complex customization and integration with existing systems. Table 1 provides a comprehensive comparison of state-of-the-art approaches.

## 2.2. Problem statement

Although several IDS based on deep learning have been suggested for improving network security, yet these methods inherently have drawbacks such as a high error rate and failure to detect cyber-attacks with relatively poor system throughput. Hence, a multi-layered architecture has been proposed that uses deep learning algorithms with high accuracy to identify malicious packets in each layer. The attackers create malicious packets based on the same features as non-malicious packets. The attack packets are separated while gathering the received packets from the users, and the attackers are identified. In the multi-layered architecture, the malicious nodes were detected in the individual layers with the help of low-level complexity-based algorithms. The objective of the proposed work is to identify and control different security assaults, including DDoS attacks.

**Table 1. Comprehensive comparison of state-of-the-art approaches.**

| Method | Benefits | Drawbacks |
|---|---|---|
| Deep Learning for SDN-Enabled IDS in IoT (Chaganti R. et al. [21]) | It utilizes LSTM for effective attack detection. Outperforms traditional ML classifiers. | Requires large labeled datasets and significant training time. Not validated in adversarial environments. |
| Decision Tree-Based IDS (Guezzaz et al. [22]) | Enhances data quality and training efficiency via pre-processing and entropy-based feature selection. Improved with DL techniques. | Prone to overfitting and limited scalability for complex attack scenarios. |
| XGBoost for Imbalanced Multi-class Classification (Le et al. [23]) | This method achieves high F1-scores (99.9%, 98.7%) for two datasets. Demonstrates strong accuracy. | Requires testing for IIoT-specific applications. Limited generalizability beyond pre-processed datasets. |
| LSTM Classifier for IoT Networks (Wani et al. [24]) | Detects unauthorized intrusions effectively in IoT networks using DL. | Needs validation in real-time environments with simulated attacks. High computational requirements. |
| SDN-Based Architecture for DDoS Mitigation (Perez-Diaz JA et al. [25]) | Successfully mitigates DDoS attacks across two network topologies using ML. | Performance could improve with advanced DL techniques. Limited evaluation criteria for comprehensive comparisons. |
| GRU-RNN for SDN (Tang TA et al. [26]) | Achieves an 89% detection rate. Handles both anomalous and legitimate traffic effectively with minimal impact on controller performance. | Requires additional features to reduce overhead and increase accuracy. Concerns of overfitting to specific datasets. |
| Real-Time IDS with RNN (Albahar MA [27]) | Combines flow collection, anomaly detection, and mitigation for real-time detection. Requires fewer features, reducing computational complexity. | Slight impact on controller performance. Scalability to larger, complex environments remains uncertain. |
| Collaborative SDN-IoT Architecture (Nguyen TG et al. [28]) | Optimizes resource management and communication overhead. Handles diverse datasets from multiple sources effectively. | Needs exploration of more cyberattack scenarios. Requires robust algorithms for handling larger datasets. |
| Efficient DDoS Defense with eHIPF (Phan TV and Park M [29]) | Improves DDoS detection speed with eHIPF and hybrid ML models. | Requires more comprehensive evaluation for comparison with other ML methods. Scalability and broader applicability need validation. |
| Multi-Level Hybrid IDS for SDN (Latah M. and Toker L. [30]) | Employs KNN, ELM, and H-ELM for reduced testing time. Multi-level hybrid approach shows promise. | Limited accuracy (84.29%) and high false alarm rate (FAR). Needs further enhancements for reliability and scalability. |
| Toony et al. [31] | Multi-block's complexity may hinder deployment on resource-constrained IoT devices. | Complex multimodule structure, Requires high computational resources for processing |
| Khedr et al. [32] | It utilizes machine learning with a reduced set of features. | Detection is limited to DDoS attacks. May not generalize well for other types of attacks. |

## 3. Proposed methodology

### 3.1. System model

This paper proposes MAC address-based authentication for SDN intrusion detection that utilizes a conditional generative adversarial network with dual discriminators and a growing self-organizing map to enhance network security and prevent cyber-attacks in wireless networks and SDN environments. The architecture of the proposed system model is depicted in Fig 1. The proposed system consists of four layers: the authentication process, infrastructure layer, control plane layer, and application layer. The key elements of the suggested IDS architecture are defined in the subsequent section, along with an explanation of their processing techniques. The 5G network has several numbers of users denoted as $A_1$, $A_2$, $A_3$, ......$A_N$. There are '$x$' number of switches and '$y$' number of controllers as $M_1$, $M_2$, ......$M_x$ and $C_1$, $C_2$, ......$C_y$ respectively. The Four-Q curve [34] is used to authenticate 5G users once they submit their credentials. As a result, the MIMA is lessened, and only authorized users are permitted. To prevent flow table overload attacks, the selection of optimal switches is crucial. After the switch selection, the packets are further classified using the DDcGAN approach and categorized at the application layer by employing the GSOM algorithm.

### 3.2 Medium access control

Each virtual network node is assigned a unique number known as its media access control (MAC) address. They identify individual nodes and facilitate data packet transmission on wired and wireless networks. Additionally, MAC addresses are utilized to allocate IP addresses to network nodes. The system model of the proposed methodology is depicted in Fig 2.

### 3.3. The authentication processes

The proposed system begins with the authentication process, where individual nodes in the network are required to verify their identities using the Four-Q curve technique. This step ensures that only authenticated users with valid MAC addresses gain access to the network. The number of 5G users has significantly increased, with diverse requirements. Initial user authentication helps prevent attackers from entering the network. To register with the 5G network administrator, users must provide the credentials such as password, identity, location, medium access control address, physically unclonable function. If attackers are aware of these security credentials, they can access the legitimate user's credentials. During the authentication process, security credentials are encrypted using the four-Q-curve approach, which overwhelms Man in the Middle Attacks (MIMA).

### 3.4. Four-Q curve approach

Four-Q is a high-performance elliptic curve that provides a 128-bit security level. It uses a four-dimensional decomposition to minimize the total number of elliptic curve group operations. It is designed for key agreement schemes and digital signatures. The curve is defined over a two-dimensional extension of the prime field defined by Mersenne prime $2^{127} - 1$. The curve is defined by a twisted Edwards equation (1)

$$F_{(P^2)}/E_C = -X^2 + Y^2 = 1 + dx^2 y^2$$

(1)

Where '$E$' denotes the Edwards curve, '$p$' represents the Mersenne prime number, and '$d$' indicates a constant value. Variables '$X$' and '$Y$' are affine with respect to the letter '$E$.' In this way, the MIMA is mitigated, and only authenticated users are allowed [35].

The proposed system uses the Four-Q curve authentication procedure to secure software-defined networks (SDNs) supporting 5G communication, which are increasingly vulnerable to cyber intrusions. This high-performance elliptic curve in cryptography offers an efficient and secure method for authenticating devices and users within the network. The system

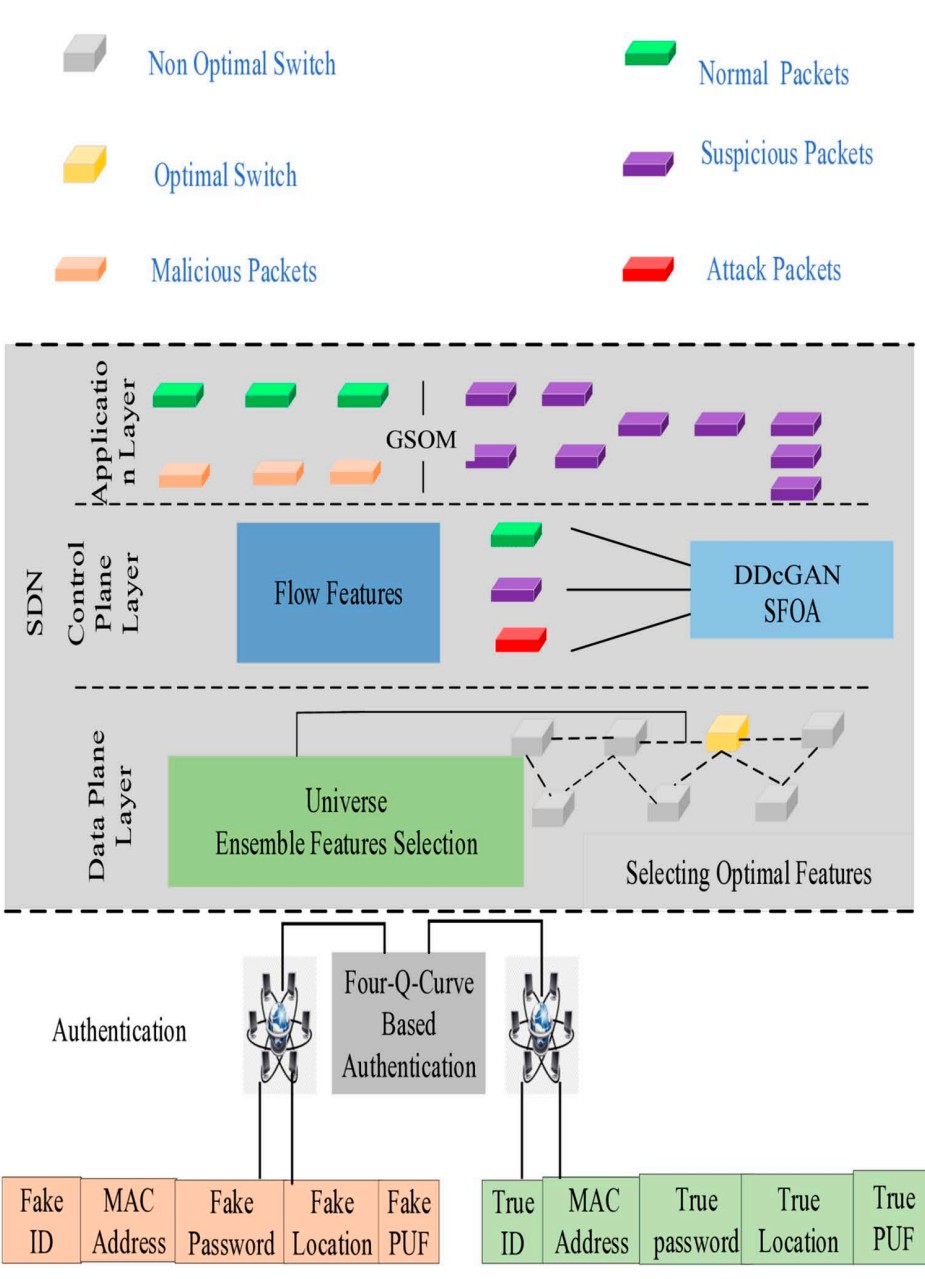

**Fig 1. Architecture of proposed system model.**

is based on MAC addresses, ensuring lightweight yet robust authentication with reduced computational overhead. This is particularly important for SDNs, where real-time processing is critical. The Four-Q curve's ability to achieve secure key exchanges with smaller key sizes and faster computations enhances the system's overall responsiveness and security. The system also establishes a secure foundation for the multilayered intrusion detection process, using deep learning techniques like DDcGAN to classify network traffic into normal, assault, and suspect categories. This ensures the integrity of the initial security layer, reducing false positives and enhancing accuracy, making the system more efficient and reliable.

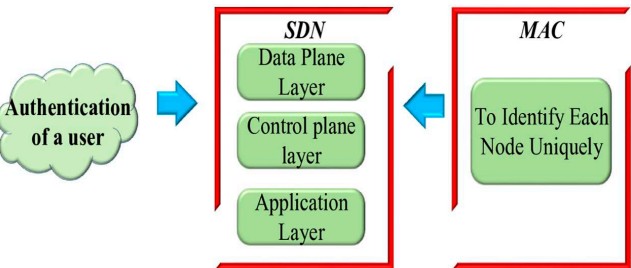

**Fig 2. System model of the proposed method.**

### 3.5. SDN architecture

The networks are capable of being intelligent, centralized controlled, and programmed by using the technology known as software defined networking. Despite fundamental technological advancements, administrators can integrate and exercise control over the entire network. The conventional architectural design description for an SDN consists of three discrete levels:

- *Infrastructure Layer*
- *Control Plane Layer*
- *Application Layer*

**3.5.1. Infrastructure layer.** The access points are responsible for sending the packets to the switches in 5G SDN networks, which receive more packets, so every incoming packet needs to be controlled. The data plane layer selects the optimal switches to route the packets to control congestion attacks. The gateway uses a univariate ensemble feature selection technique to select the best switch.

- *Univariate analysis and ensemble-based feature selection for switch optimization*

Feature selection process improves accuracy and predictability by selecting critical variables and eliminating redundant ones. In this context, univariate analysis is a simple method for analysing data with only one variable, aiming to describe and identify patterns within the data. This technique involves two stages: prioritizing features in the unified feature scoring stage and establishing a cutoff point in the threshold value selection stage to select the most important features. Thus, deep learning models get trained with more robustness. The schematic representation of univariate ensemble-based feature selection is shown Fig 3.

**3.5.2. Control plane layer.** The flow characteristics of the data packets are recovered at the control plane layer, where DDcGAN is used to categorize them into attack, normal, and suspect packets.

- **Classification using DDcGAN**

The DDcGAN model analyzes the extracted features from the packet flow to classify data into attack, normal, and suspicious categories. It consists of both local and global discriminators, along with a generator. The conventional GAN architecture consists of a discriminator and generator, where the generator endeavors to generate persuasive data packets to mislead the discriminator. The discriminator systems must select either low-level or high-level features as fundamental restrictions. The input and selected features for data packet transmission are presented in S1 Table.

- **Generator**

The generator uses a convolution design with a max pooling layer to down-sample and up-sample the packet. It employs a stride of '2' and two '3x3' convolution layers for down-sampling. The up-sampling design resembles the method of

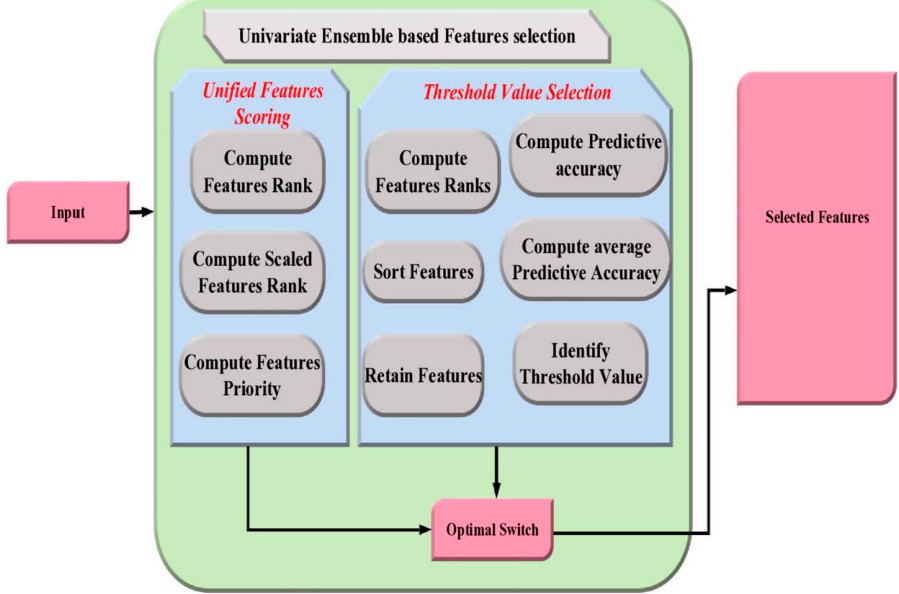

**Fig 3. Schematic representation of univariate ensemble-based feature selection.**

down-sampling but replaces the max pooling layer with a convolution layer. The problem can be addressed using a generator focus loss algorithm.

• **Discriminator**

The DDcGAN [36] system model employs both local and global discriminators. The local loss function of these discriminators is known as a binary cross-entropy (BCE) loss. The global discriminator has fewer layers because it is designed to process packets with fewer ground truth characteristics. The local discriminator consists of a leaky ReLU activation function, two batch normalization layers, and five convolutional layers. The discriminator system includes four fully connected layers. The aggregate GAN loss value is crucial for training the model. The local loss and BCE loss of the discriminator are considered, and they are represented in equation (2),

$$T_{LOSS} = \eta(G_{LOSS} + \vartheta L_{LOSS}) + \lambda Gen_{LOSS} \qquad (2)$$

In equation 2, '$T_{LOSS}$' represents the total loss used to optimize the model during training. The term '$G_{LOSS}$' denotes the global loss, which measures the performance of the global discriminator in distinguishing between real and generated data. Similarly, '$L_{LOSS}$' represents the local loss, derived from the local discriminator, which focuses on specific regions of the input for a more detailed analysis. The generator loss is indicated by '$Gen_{LOSS}$,' which guides the generator to produce samples that the discriminators classify as real, thereby enhancing the quality of the generated outputs. The parameter 'η' is a hyperparameter that balances the contributions of the global and local losses to the total loss. When 'ϑ>1', the local loss becomes more significant than the global loss, emphasizing the role of the local discriminator in the training process. The scaling factor 'λ' determines the weight of the generator loss, and when 'λ' is greater than other parameter values, it indicates that the generator is in a dominant training mode. Optimizing the generator loss '$Gen_{LOSS}$' by adjusting 'λ' enhances the accuracy and reduces the error rate of the DDcGAN, leading to improved overall performance.

The dual discriminator architecture in the DDcGAN model integrates local and global discriminators to enhance intrusion detection accuracy by leveraging their complementary strengths. The local discriminator focuses on identifying

optimal features of packet data, particularly those with detailed ground-truth characteristics. It incorporates a Leaky ReLU activation function, batch normalization layers, and convolutional layers, which work together to extract intricate patterns and relationships from the input data. In contrast, the global discriminator processes high-level features and operates on packets with fewer ground-truth characteristics, utilizing a simplified architecture with fewer layers for efficient computation. The model uses BCE loss to evaluate how well the discriminators distinguish real packets from generated ones. To improve training, the model employs an aggregate loss function '$T_{LOSS}$' that balances the contributions of local and global losses '$L_{LOSS}$' and the generator loss '$G_{LOSS}$.' By dynamically adjusting these components and optimizing '$Gen_{LOSS}$' with the SFOA, the dual discriminator system achieves enhanced feature representation, resulting in improved accuracy for classifying regular, suspicious, and attack packets [37]. Algorithm 1 provides the pseudocode for the DDcGAN algorithm.

---

**Algorithm 1.** DDcGAN Algorithm for Classifying Data Packets

---

**Input** Extracted features
**Output** Synthetic packet flow data
**Initialize** DDcGAN model
Generator network, discriminator network
$(d_{local},\ d_{local})$
**Define** parameters
Learning rate η, weight for local discriminator lossλ
Generate fake packets z using generator
**Calculate** loss for $d_{global}$
**Calculate** loss for $d_{local}$
Total loss of discriminator:
$T_{LOSS} = \eta(G_{LOSS} + \vartheta L_{LOSS}) + \lambda Gen_{LOSS}$
**Calculate** generator loss using discriminator feedback
Total generator loss: $G_{LOSS}$
**Update** generator weights
**Repeat** until max iterations
**End**

---

### 3.5.3. DDcGAN optimization using sheep flock optimization algorithm.

The global loss parameter '$G_{LOSS}$' of the DDcGAN model, as mentioned in the previous section, minimized using SFOA. The SFOA is chosen for optimizing the DDcGAN model in intrusion detection due to its ability to balance exploration and exploitation in complex, high-dimensional problem spaces. Unlike traditional optimization methods, SFOA leverages a population-based search inspired by the natural behaviour of sheep, which helps avoid local minima and provides a global perspective on the solution space. Its adaptability to dynamic, evolving problems makes it ideal for applications like intrusion detection, where data distributions can change over time. Moreover, SFOA's robustness to noise and outliers, along with its efficiency in handling multi-objective optimization, allows it to effectively minimize the DDcGAN's global loss, outperforming alternative strategies in terms of both convergence speed and solution quality. The algorithm's intuitive, nature-inspired mechanics make it an ideal choice for optimizing the intricate parameters of DDcGAN's in intrusion detection systems.

*Step 1. The Initialization*

In the initial stage, the sheep flock is optimized in terms of its population size, maximal iterations, measurements, and associated cost functions. The proposed method optimizes SFOA parameters using benchmarking dataset operations, thereby enhancing the overall effectiveness of the optimization procedure. Additionally, the weight parameters and their associated variables are initialized for achieving the optimal solution, as specified in equation (3).

$$S = int\left(N, D_{avg}(k,m), R(k,m)\right)$$

(3)

where '$R^{K \times M}$' designates the probability matrix and '$D_{avg}^{K \times M}$' represents the average expected outcome of the 'N' data.

### Step 2. Randomized Generation

After the initialization process, the randomly assigned positions for each member are established. Due to the segmentation of the population, individuals can be allocated to either the sheep or goat groups. To determine the optimal value, the random placement values are compared to previous values. Using equation (4), the quantity that contributed to the initial position of an operation is determined.

$$P_x = \left(2 * R_{g(S,G)}\right) * rand - R_{g(S,G)} \tag{4}$$

where '$R_{g(S,G)}$' is the grazing radius for sheep '($S$)' and goat '($G$),' and '$rand$' is a random number between 0 and 1. Thus, using mathematical equations (5) and (6), we can determine the grazing radius for both sheep (represented by '($S$)') and goats (represented by '($G$)').

$$R_{g(S)} = 0.001 * (U_B - L_B) * T \tag{5}$$

$$R_{g(G)} = 0.1 * (U_B - L_B) * T \tag{6}$$

where '$U_B$' is the upper limit, '$L_B$' is the lower limit, and '$T$' is the total number of repetitions indicated by equation (7).

$$T = 1 - [iteration\_no / max\_iteration] \tag{7}$$

### Step 3. Determining the Fitness Function

The fitness function has been formulated to facilitate the identification of optimal problem solutions. Using equation (8), the global loss variable of the DCGAN algorithm was optimized.

$$FitnessFunction = Minimize\left(G_{loss}\right) \tag{8}$$

where '$G_{LOSS}$' represents the total DDcGAN loss.

### Step 4. Explore a new location

If the new site has an improved cost value as compared with the previous location, the sheep will relocate there. The move section helps the shepherd determine the best course of action, helping the flock find a new home. The shepherd's order, which is estimated using equation (9), determines the optimal site based on the resulting movement.

$$V_{Sp,1} = (1 - T)C.rand(1, Dim)(X_{gbest} - X) \tag{9}$$

where '$X$' represents the current sheep location, '$X_{gbest}$' the ideal fitness value, and $(1, Dim)$ is an array of random values between 0 and 1. '$C = 3 * rand$' and '$V'_{S_{p,1}}$' denotes the sequence of movements of the shepherd depending on instances of movement. In addition, equation (10) can be used to compute the motion caused by the sheep's desire to repeat their most memorable prior experience.

$$V_{Lbest,1} = C.rand(1, Dim)(X_{Lbest} - X) \tag{10}$$

where '$X_{Lbest}$' symbolizes the optimal fitness quantity, '$(1, Dim)$' denotes a randomly generated array that is within the two extremes [0, 1], and '$V_{Lbest,1}$' indicates the sheep's desire based on its preceding finest moment. Additionally, using equation (11), the movement caused by the sheep's fascination with approaching other sheep is estimated.

$$V_{other,1} = C.rand(1, Dim)(X_{rand.S} - X) \tag{11}$$

Where '$X_{rand.S}$' indicates the unpredictability of the sheep's position and '$V_{other,1}$' represents the sheep's inclination to approach other sheep considering the total number of repetitions '$T > 0.3$.'

***Step 5. Parameter optimization by updating the location***

Depending on the movement of the sheep, the measure of herd dispersion is decreased to '$T \leq 0.3$.' Equation (12) is used to determine the pathway that arises from the shepherd's instructions to the most appropriate location.

$$V_{Sp2,1} = C_1(1 - T)(X_{gbest} - X) \tag{12}$$

Further, equation (13) is used to calculate the movement caused by the sheep's fixation on its past favourable experience.

$$V_{m,1} = V \begin{cases} V_{Sp1,1} + V_{Lbest,1} + V_{other,1} & T > 0.3 \\ V_{Sp2,1} + V_{Lbest,1} & T \leq 0.3 \end{cases} \tag{13}$$

where '$V_{m,1}$' is the predicted velocity determined by sheep movement. Furthermore, using equation (14), the sheep's present location is replaced with its future location.

$$X_{Iteration+1} = V(X_{Iteartion} + V_{M,1}) \tag{14}$$

The velocity of the sheep's movement is then used to update the sheep's position. To acquire the best solution by optimizing the parameters, the value of '$V_{m,1}$' with the fitness value '$V$' is utilized. Therefore, the model's optimal value has been reached. The flowchart of the sheep flock optimization algorithm is depicted in S1 Fig.

***Step 6: Termination***

The sheep flock optimization [38] method reduces the DDcGAN error rate by optimizing the global loss parameter, resulting in a decreased error rate.

To summarize the above findings, the key steps of SFOA include updating factors such as random position assignment and fitness evaluation based on parameters like grazing radius and movement speed. These iterative updates reduce the error rate and optimize the generator's performance, allowing the DDcGAN to identify attack packets more accurately and enhancing the overall intrusion detection accuracy by minimizing the '$G_{LOSS}$.' The SFOA pseudocode is provided in Algorithm 2.

---

**Algorithm 2.** Pseudocode of the SFOA algorithm

---

*Input:* $Pop_{SIZE}, Pro_{SIZE,} U_B, L_B, Max_{ITERATION},$
*output:* $gbest$
*update:* $v_{MAX}$
*initialize the sheep population*$(U_B, L_B)$
*iteration*=$iteration \leq Max_{ITERATION} * 0.1$
*iteration*=$iteration \leq Max_{ITERATION} * 0.1$
$FitnessFunction = Optimize(G_{loss})$
*Update* $T$
*For*$i = 1$ *to*$Pop_{SIZE}$ *do*
*If (Grazing Condition)*
*Update*$P_x$
$x(j) = sheep(i) \cdot x + P_x$
*if cost* $(x(j)) < sheep.cost$
$sheep(i) \cdot x = sheep(i) \cdot x + P_x$
*Update* $gbest, Lbestby(sheep(i))$
*End*
*Else*
*Update* $C_1, V_{Sp\ 1,1}, V_{Sp\ 1,2}, V_{Lbest}, V_{other}$

```
Update  V_M
Calculate_  sheep position &cost(sheep(i) , V_M))
Update  gbest, Lbestby (sheep(i))
End
End
ITERATION+ = 1
End
Return  gbest
```

**3.5.4. Application layer.** At the application layer, suspicious packets are further classified into normal and malevolent utilizing a growing self-organized map (GSOM). This procedure detects DDoS attacks.

• **Classification of suspicious packets using GSOM**

An unsupervised neural network of the GSOM type is based on the SOM methodology. Following are the three steps of the GSOM [39] algorithm: The GSOM algorithm for categorizing suspicious packets has been presented in S1 Algorithm.

• *Initialization step*

• *Growing period*

• *Smoothing stage*

*Step1. Initialization*

In the initialization step, the random integers are used to configure the initial packet weight vectors. The growth threshold (GT) is computed using equation (15).

$$GT = -D * In(SF)$$ 

(15)

where *'D'* represents the dimensions of data and *'SF'* indicates the spread factor. The network continuously receives data packets input during growth phase. The winning packet is selected based on the Euclidean distance between its weight vector and the input vector. The weight vectors of the winner and its neighbouring packets are adjusted by equation (16),

$$W_j(K+1) = \begin{cases} W_j(K), j \neq N_{K+1} \\ W_j(K)LR(K) * (x_K - W_j(K)), j \in N_{K+1} \end{cases}$$

(16)

where '$W_j$' is the winning packet's weight vector, '$j$,' '$K$' is the current time, '$LR$' is the learning rate, and '$N$' is the winning packet's neighbourhood.

*Step 2. Growing period*

At the boundary rate, new packets are produced when the cumulative error exceeds the growth threshold. These new packets weights are adjusted to match the weights of adjacent packets. Errors found in non-boundary packets are distributed among neighbouring packets. The growing phase repeats until all input has been provided and packet growth is minimized.

*Step 3. Smoothing stage*

The smoothing step slows down the learning rate by classifying suspicious packets as either normal or malicious. Then it performs weight adaptation in a smaller neighbourhood and detects DDoS attacks. Fig 4. illustrates the flow diagram of the proposed methodology.

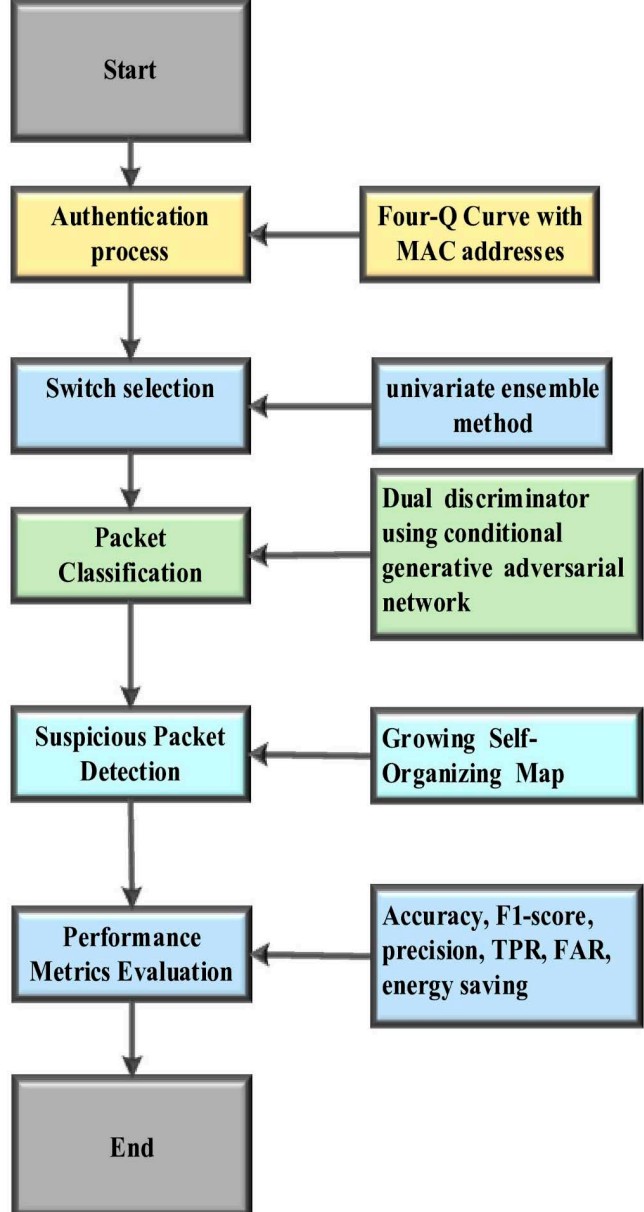

**Fig 4. Flow diagram for the proposed methodology.**

## 4. Results and discussion

### 4.1. Simulation setup

The proposed work has been implemented using the network simulator (NS-3) software. The simulation is performed over 300 nodes. The simulation parameters of the suggested DDcGAN and SDN network are given in Tables 2 and 3.

### 4.2. Performance measures

Table 4 lists the analytical expressions used to calculate the various performance metrics, such as accuracy, F1 score, precision, delay, energy consumption, throughput, specificity, true positive rate, and false alarm rate. These expressions

**Table 2. Simulation parameters of DDcGAN.**

| Specifications | Values |
|---|---|
| *Input parameters* | 40 |
| *Hidden layers* | 3 |
| *Neurons in hidden layer* | 40 |
| *Trainable parameters* | 4240 |
| *Activation function of hidden layer* | ReLU |
| *Activation function of output layer* | Soft Max |
| *Cost function* | Sparse Categorial Cross entropy |
| *Dropout Rate* | 0.2 |
| *Learning Rate* | 0.01 |
| *Epoch* | 500 |
| *Batch size* | 500 |

**Table 3. Simulation parameters of SDN network.**

| SDN network | OpenFlow |
|---|---|
| *Total nodes* | 300 |
| *Simulated region* | 300∗300 sq. Meters |
| *Data rate* | 8 Kbps |
| *Routing Protocol* | AODV |
| *Carrier frequency* | 2.4 GHz |
| *Bandwidth* | 10 MHz |
| *Node Placement* | Uniform Random |
| *Propagation Delay* | 15 ms |
| *Simulation Time* | 60 seconds |
| *Node Velocity* | 0 m/s |

**Table 4. Performance metrics.**

| Performance Measures | Formulae |
|---|---|
| Accuracy | $Accuracy = \frac{TRUE_{POSITIVE} + TRUE_{NEGATIVE}}{TRUE_{POSITIVE} + TRUE_{NEGATIVE} + FALSE_{POSITIVE} + FALSE_{NEGATIVE}}$ |
| F-Measure | $F1 - score = 2 * \frac{P*R}{P+R}$ |
| Precision | $Precision = \frac{FALSE_{POSITIVE}}{TRUE_{POSITIVE} + FALSE_{POSITIVE}}$ |
| Delay | $Delay = (packet sending time - packet receivng time)$ |
| Energy Consumption | $Energy\ Consumption = \frac{\sum Energy\ utilized\ in\ each\ node}{Initial_{energy}}$ |
| Throughput | $Throughput = \frac{\sum Packets\ reaching\ at\ the\ estimation}{Time}$ |
| True positive rate | $TruePositiverate = \frac{TRUE_{POSITIVE}}{TRUE_{POSITIVE} + FALSE_{NEGATIVE}}$ |
| False alarm rate | $FalseAlarmRate = \frac{FALSE_{NEGATIVE}}{FALSE_{NEGATIVE} + TRUE_{POSITIVE}}$ |
| Specificity | $Specificity = \frac{TRUE_{NEGATIVE}}{TRUE_{NEGATIVE} + FALSE_{POSITIVE}} * 100\%$ |

are essential in establishing the worth of the DDcGAN method in context to the existing algorithms such as GRU-RNN, MLP, and RNN-SDR by a significant margin.

## 4.3. Performance analysis

This section shows the performance analysis of the proposed DDcGAN method. To demonstrate the efficiency of the proposed method, its performance is compared with state-of-the-art methods such as GRU-RNN, MLP, and RNN-SDR, respectively. Fig 5. displays the network simulation diagram of the suggested method, with red colour dots indicating randomly distributed nodes across the network. The green colour lines show the interconnection amongst the nodes. The DDcGAN algorithm is applied to this network simulation to classify malicious and suspicious packets with the help of authentication. The subsequent sections discuss how the DDcGAN method accurately detects intrusions in the network.

Fig 6 illustrates the accuracy performance of the DDcGAN model. Evaluating accuracy is vital when assessing an intruder's overall prediction capability. Higher levels of accuracy signify an improved ability to detect network intruders, especially as the number of attackers increases. Categorically, using the DDcGAN algorithm with 10% malicious nodes, the detection accuracy can improve in the range between 6.5% to a maximum of 7.3% compared to the existing techniques. With a more frequent percentage of malicious nodes (50%), the detection rate using DDcGAN reaches 98.29%, which is 4% higher than the closest competitor, RNN-SDR. The significant increase in the detection accuracy leads to an impressive performance in terms of a higher F1 score and enhanced precision compared to existing approaches, with the proposed method achieving an F1 score of 0.975 and precision of 95.8%, as illustrated in Figs 7 and 8.

The accurate detection and precise estimation should be substantiated with a crucial analysis of the delay in detecting the malicious nodes. Fig 9. depicts the DDcGAN delay analysis, which shows a lower delay than existing techniques. The proposed DDcGAN algorithm surpasses the state-of-the-art techniques for both energy consumption and delay in networks with a higher percentage of malicious nodes. A substantial drop in delay of around 17% for lower malicious node count 10% to 33% in higher malicious nodes (50%) can be observed clearly from the figures. In fact, in an interesting revelation, the result suggests that with more malicious nodes, the system favours a higher detection accuracy with minimum delay compared to the existing methods. To further investigate the performance of DDcGAN on whether it provides good results at the cost of increased energy requirements, an analysis was performed. Even in that, as shown in

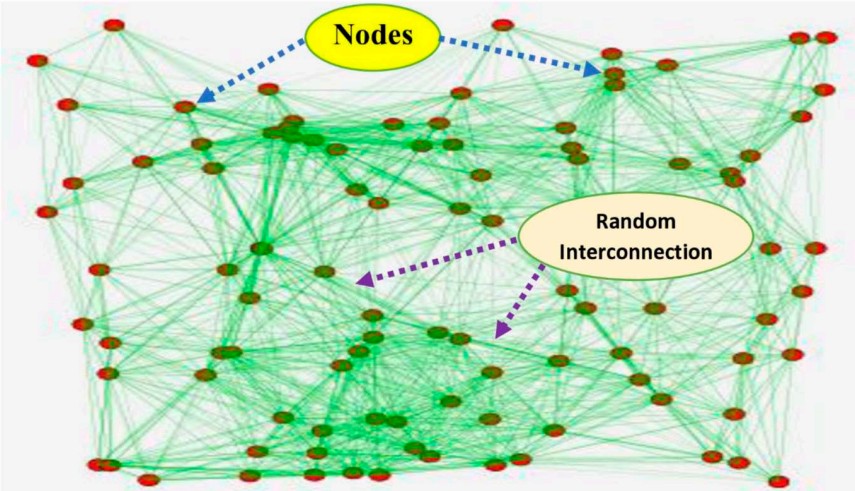

**Fig 5. Network simulation of the proposed method.**

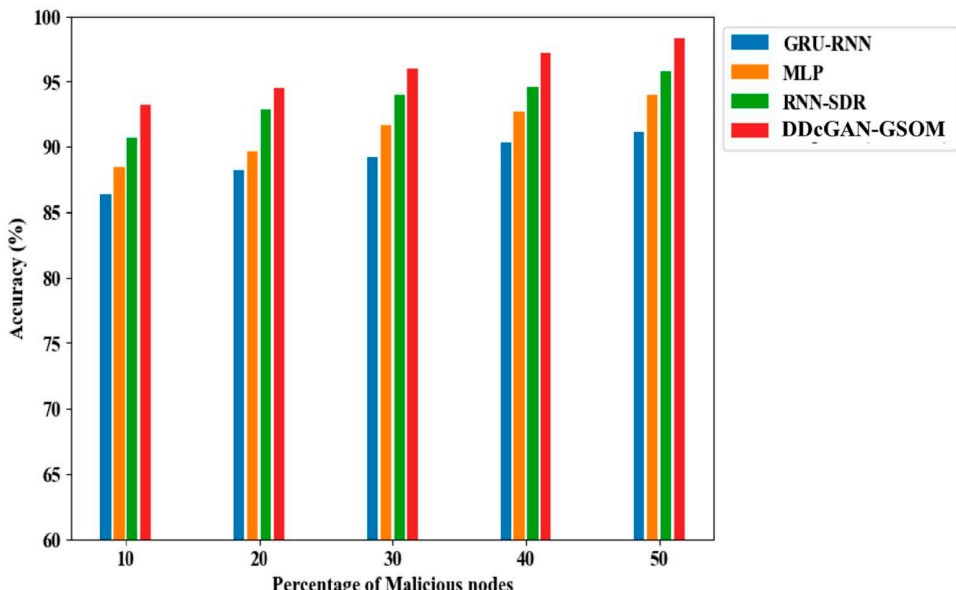

**Fig 6. Accuracy analysis of DDcGAN-GSOM with existing techniques.**

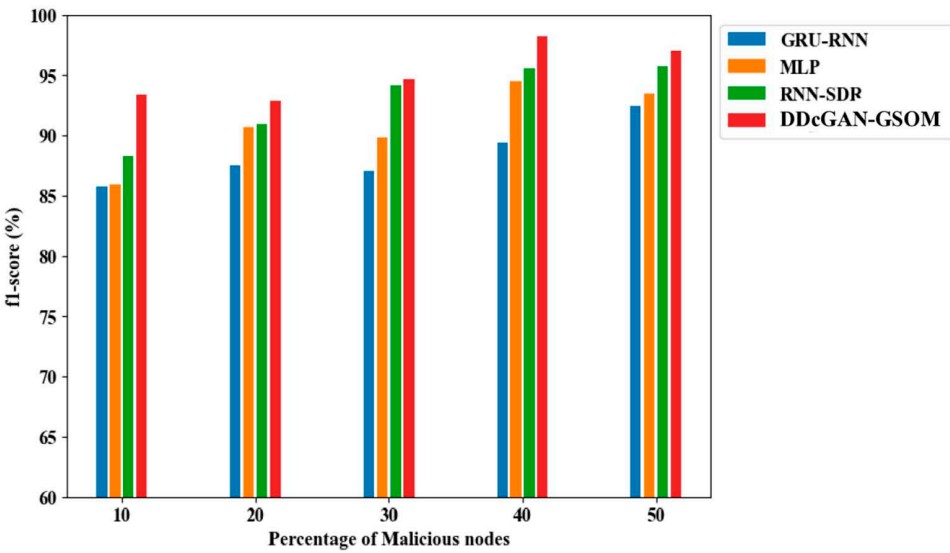

**Fig 7. F1-score analysis of DDcGAN-GSOM with existing methods.**

Figs 10 and 11, energy consumption by DDcGAN is suggestively less than the current approaches. For the 50% malicious nodes, there is a 4.5% energy savings as compared to the existing methods, which is a huge benefit. This is followed by a throughput analysis to authenticate the claim of a better showcase of DDcGAN. This is essential, as with high accuracy, a better F1 score, superior precision performance, reduced energy consumption, and lower delay, there should not be any compromise with the throughput. Fig 12. illustrates the DDcGAN throughput analysis. The total number of packets transmitted from the source to the destination in a predetermined amount of time is adjudicated for throughput analysis.

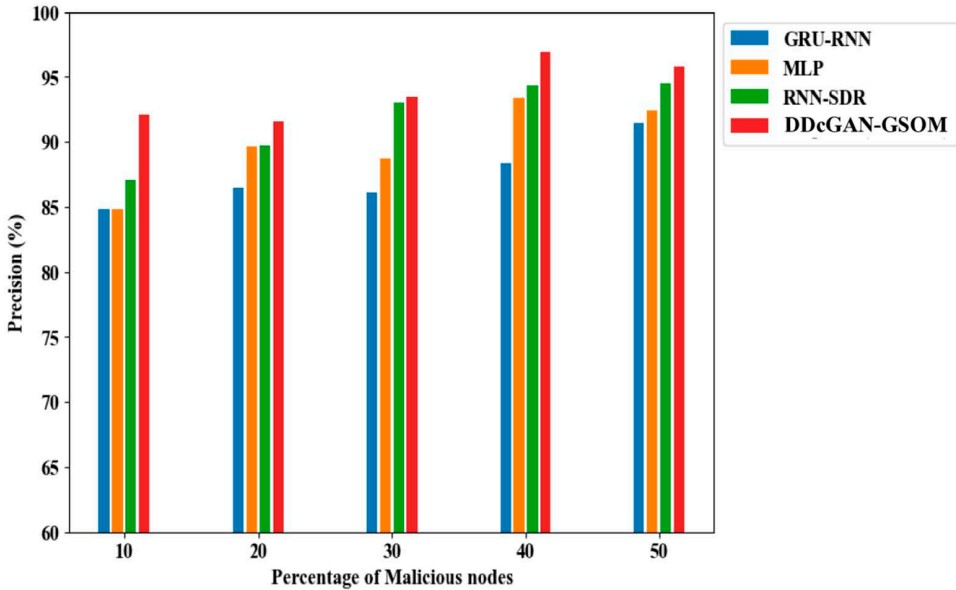

**Fig 8. Precision analysis of DDcGAN-GSOM with current approaches.**

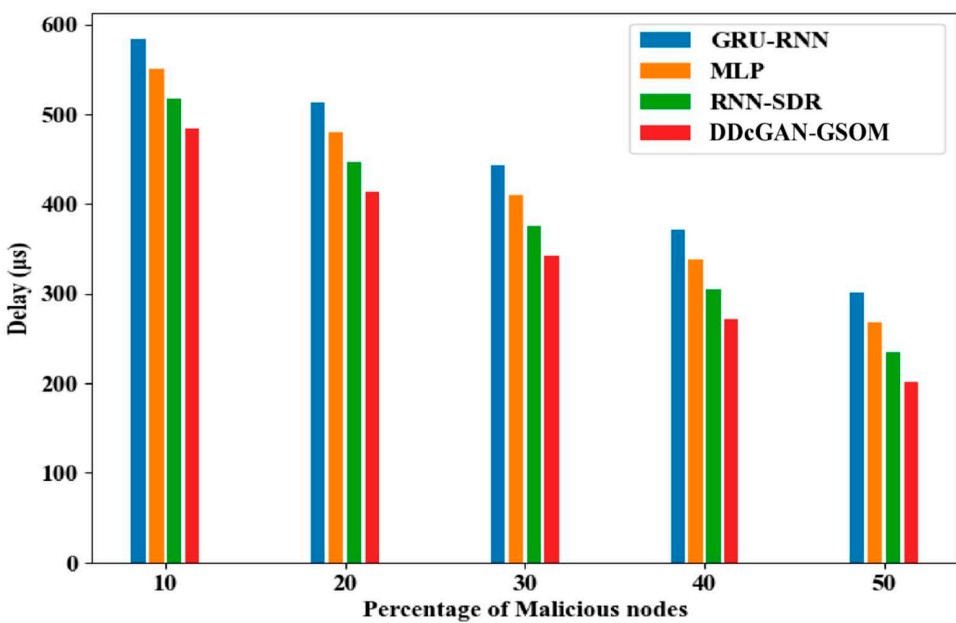

**Fig 9. Analysis of DDcGAN-GSOM delay with existing techniques.**

The DDcGAN algorithm outperforms the prevailing methods concerning throughput, true positive rate, and specificity for networks with more malicious nodes.

Now, after establishing the worth of DDcGAN, the classification parameters are needed to show similar performances. To continue further, a True Positive Rate (TPR) analysis is performed. TPR analysis of DDcGAN shows the probability that a true positive result will appear on the test more frequently and accurately. As shown clearly in Fig 13, the TPR of DDcGAN

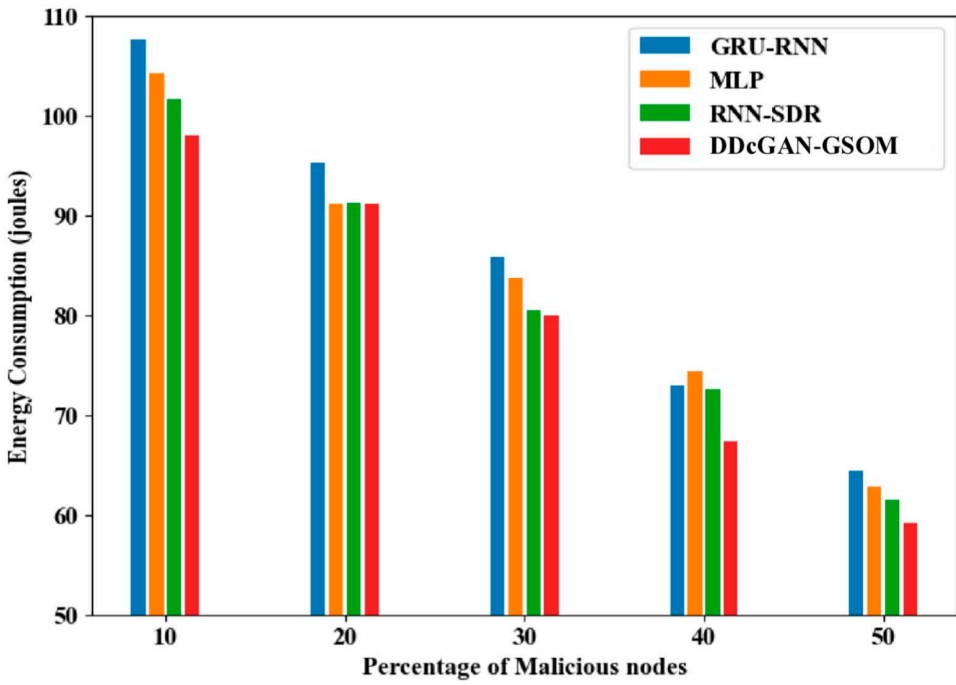

**Fig 10. Energy consumption analysis of DDcGAN-GSOM with existing methods.**

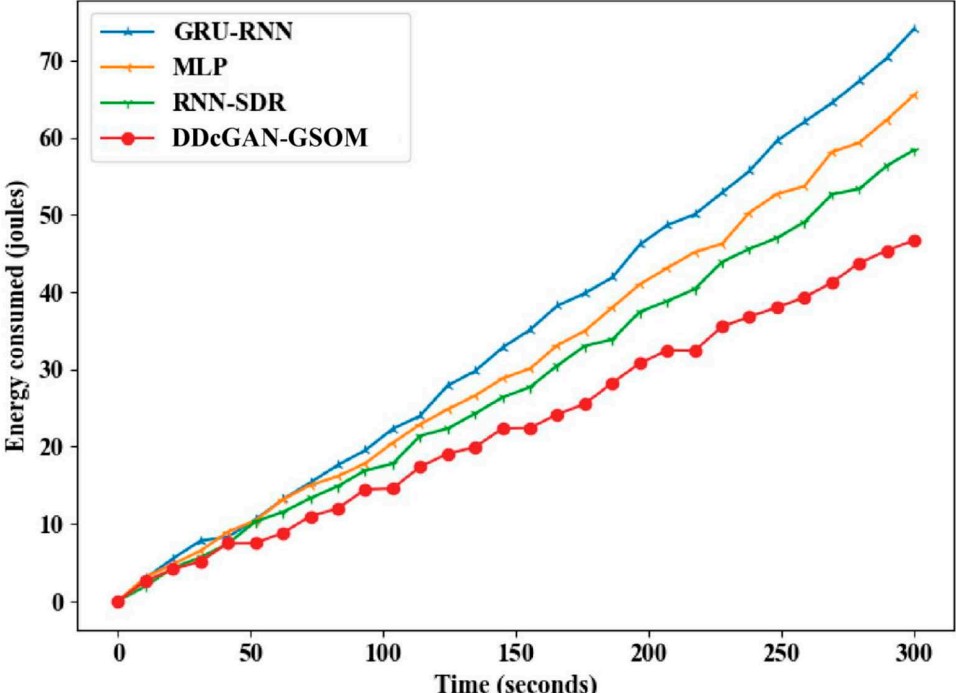

**Fig 11. Energy consumption analysis of DDcGAN-GSOM with existing approaches.**

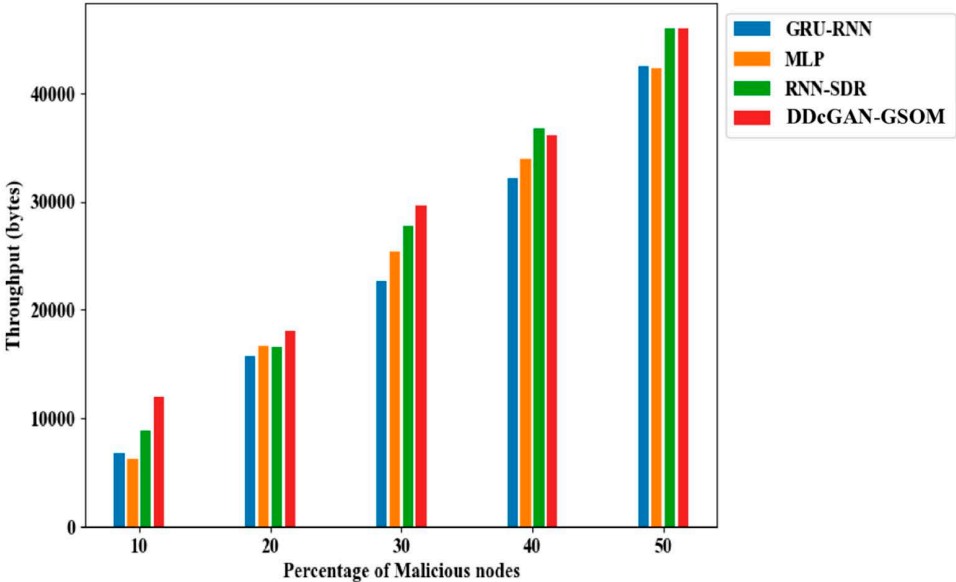

**Fig 12. Throughput analysis of DDcGAN-GSOM with prevailing approaches.**

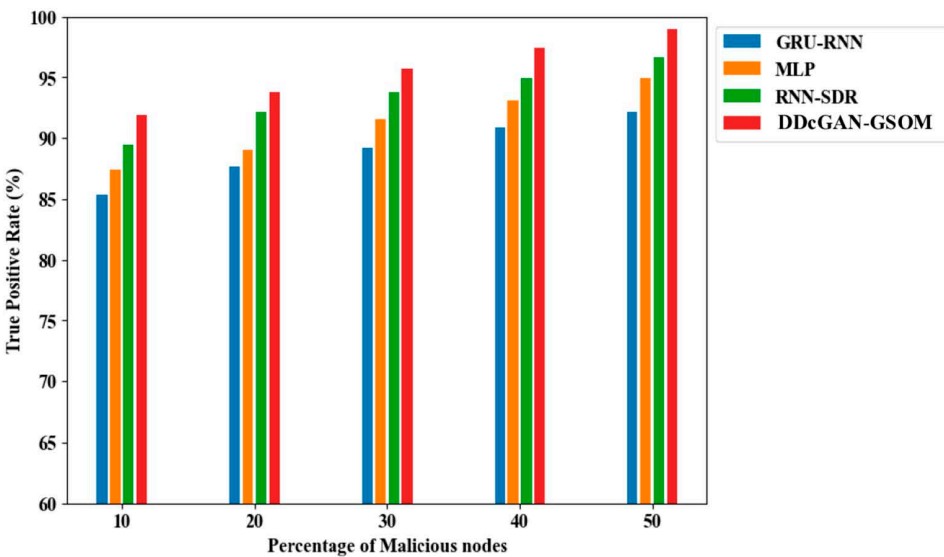

**Fig 13. True positive rate (TPR) analysis of DDcGAN-GSOM with existing techniques.**

is higher than the existing approaches by a significant amount. In fact, when subjected to a high percentage (50%) of malicious nodes, the DDcGAN attains almost 99.04% TPR. Similarly, in Fig 14., the False Alarm Rate (FAR) evaluation of DDcGAN shows that it attains far less FAR, i.e., low false detection, compared to existing techniques. Correspondingly, exposed to a high percentage (50%) of malicious nodes, DDcGAN attains a lower FAR of 2.05%. To authenticate the effectiveness of the test conducted for the scenario with malicious nodes, the specificity parameter is analyzed. The analysis shows that the proposed method attains higher specificity than the existing approaches, which is illustrated Fig 15. When

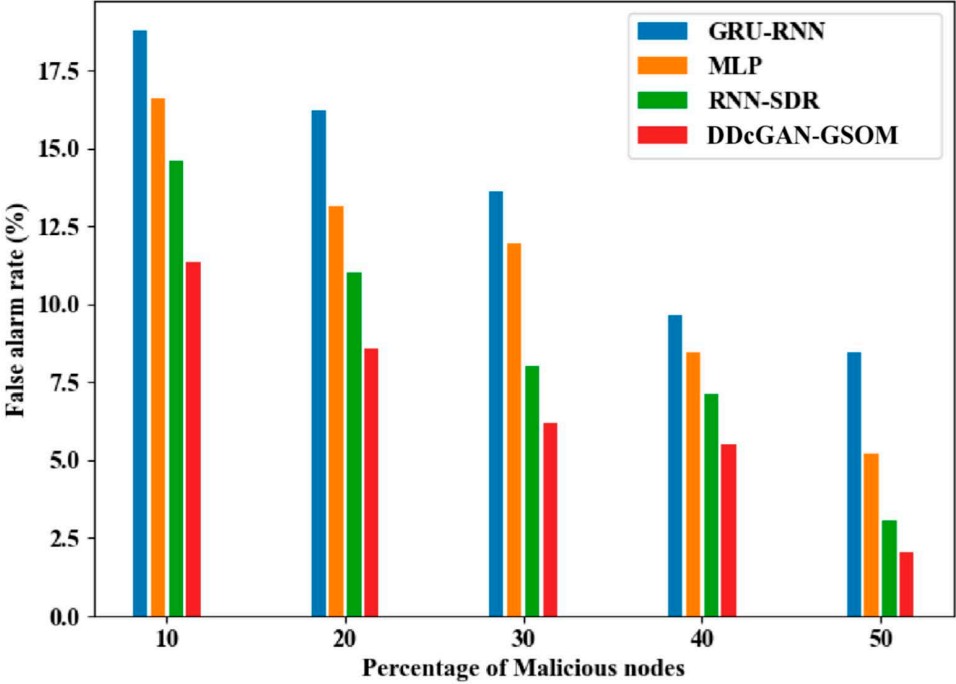

**Fig 14. False alarm rate analysis of DDcGAN-GSOM with existing approaches.**

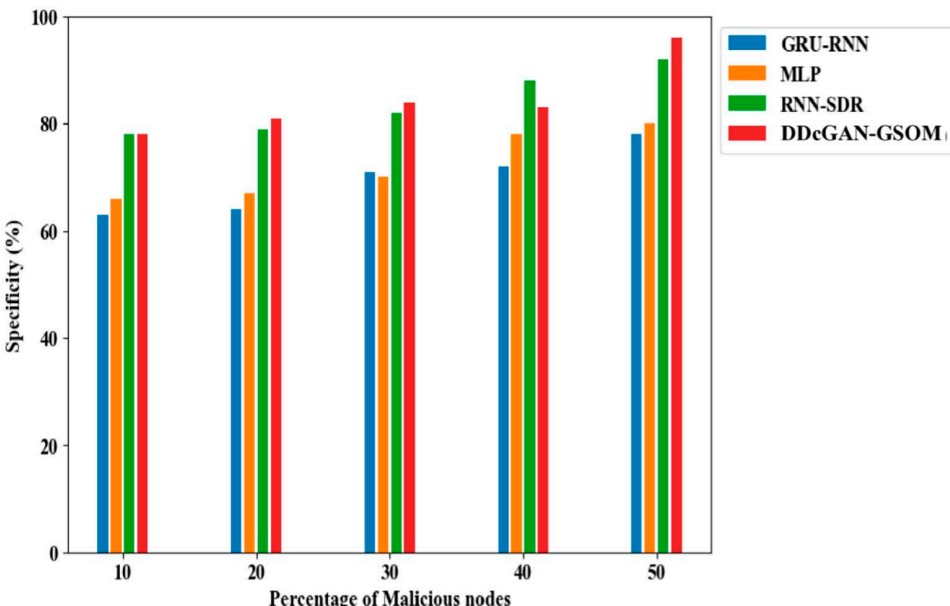

**Fig 15. Specificity comparison of DDcGAN-GSOM with prevailing methods.**

the percentage of malicious nodes is 40, the specificity of DDcGAN is slightly lower than that of RNN-SDR. But when the percentage of malicious nodes is 50, the DDcGAN attains 97% specificity, which is higher than the existing approaches.

The successful results, as discussed previously, depend heavily on how effectively the generated data deceives the discriminator into classifying it as real. A lower generator loss indicates that the generator is producing highly realistic data. Even the discriminator loss that evaluates how accurately the discriminator differentiates between real and generated data becomes an essential testing parameter. A lower discriminator loss reflects the discriminator's proficiency in detecting counterfeit data. Based on this approach, Figs 16 and 17 represent the discriminator loss and generator loss of the proposed DDcGAN method over iterations. Fig 16., illustrates a decreasing trend in discriminator loss, which starts at approximately 0.9 and stabilizes around 0.2 after 500 iterations. This trend indicates the discriminator's improved ability to distinguish between real and generated samples as training progresses. Fig 15., shows the generator loss, which begins at approximately −1.0 and increases steadily, stabilizing near −0.3 after 500 iterations. This suggests that the generator effectively adapts to improve the quality of its outputs, forcing the discriminator to work harder to differentiate between real and generated data. After analysing these loss trends, it is evident that both the generator and discriminator converge effectively, ensuring balanced adversarial learning.

To summarize the findings from the previous discussion, the DDcGAN method outperforms traditional approaches (GRU-RNN, MLP, RNN-SDR) in network intrusion detection, achieving 98.29% accuracy, a 0.975 F1 score, and 95.8% precision, which is tabulated in Table 5. It excels in efficiency, reducing energy usage by 4.5%, enhancing throughput, and achieving 99.04% TPR and 2.05% FAR. Its specificity (97%) ensures superior malicious node detection, which is listed in Tables 6 and 7. Accordingly, all these significant advantages set up the proposed system to be used in practical applications such as real-time device validation in SDNs, preventing unauthorized access while maintaining optimal performance in large-scale networks with enhanced security. It rapidly detects and prevents cyber threats, ensuring a secure network by identifying attacks like DDoS and man-in-the-middle. It reduces false positives, enhancing detection accuracy and allowing administrators to focus on real threats without unnecessary alarms. The proposed system optimizes security with minimal computational overhead, ensuring network speed, efficiency, and low power consumption in resource-constrained

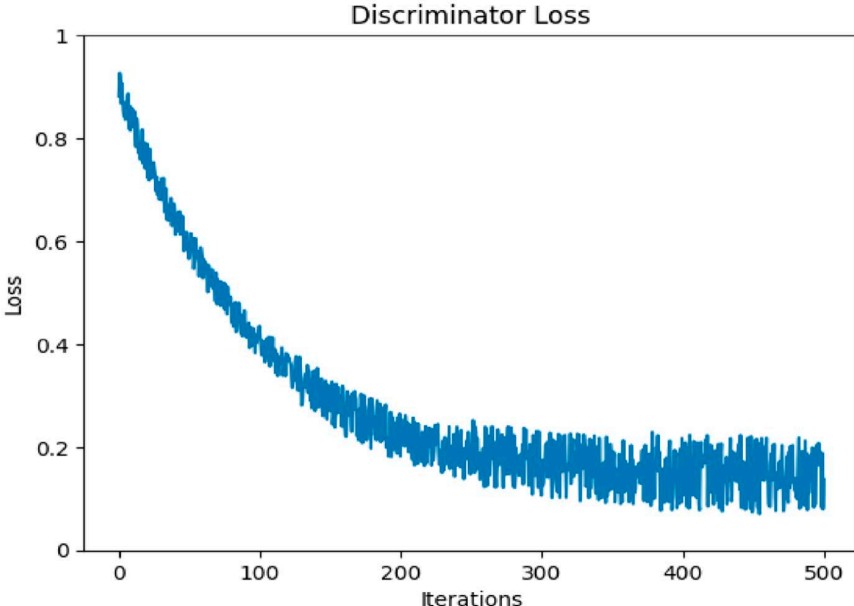

**Fig 16. Discriminator loss of suggested technique.**

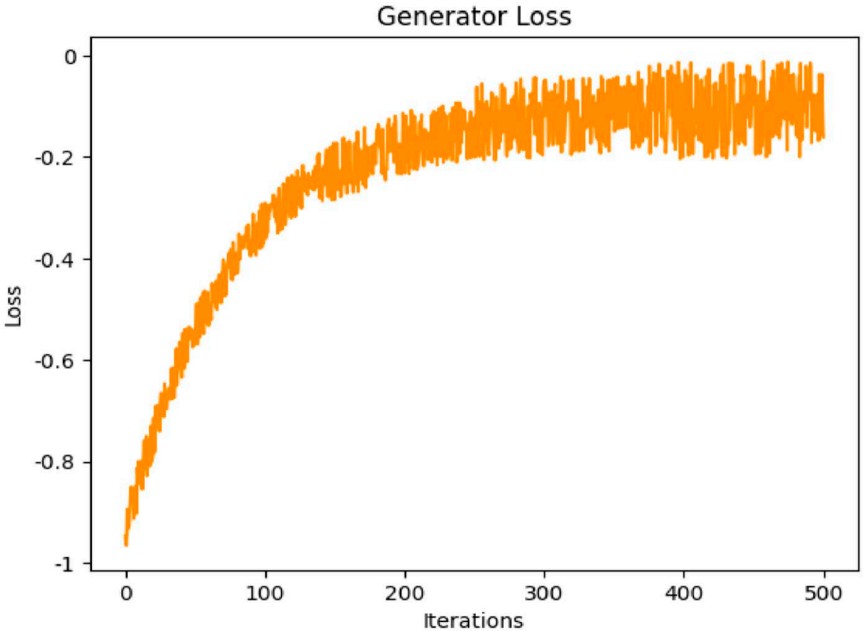

**Fig 17. Generator loss of the proposed method.**

**Table 5. Comparative analysis of DDcGAN-GSOM's Accuracy, Precision, and F1-score.**

| Methods | Malicious Nodes (10%) | Malicious Nodes (20%) | Malicious Nodes (30%) | Malicious Nodes (40%) | Malicious Nodes (50%) |
|---|---|---|---|---|---|
| **GRU-RNN** | Accuracy: 86.37% | Accuracy: 88.20% | Accuracy: 89.23% | Accuracy: 90.35% | Accuracy: 91.16% |
| | Precision: 84.8% | Precision: 86.48% | Precision: 86.1% | Precision: 88.41% | Precision: 91.46% |
| | F1 score: 0.858 | F1 score: 0.874 | F1 score: 0.871 | F1 score: 0.894 | F1 score: 0.924 |
| **MLP** | Accuracy: 88.49% | Accuracy: 89.64% | Accuracy: 91.70% | Accuracy: 92.68% | Accuracy: 94.04% |
| | Precision: 84.85% | Precision: 89.65% | Precision: 88.72% | Precision: 93.44% | Precision: 92.43% |
| | F1 score: 0.859 | F1score: 0.907 | F1 score: 0.898 | F1 score: 0.945 | F1 score: 0.935 |
| **RNN-SDR** | Accuracy: 90.69% | Accuracy: 92.89% | Accuracy: 94.00% | Accuracy: 94.63% | Accuracy: 95.85% |
| | Precision: 87.12% | Precision: 89.77% | Precision: 93.02% | Precision: 94.36% | Precision: 94.55% |
| | F1 score: 0.882 | F1 score: 0.909 | F1 score: 0.941 | F1 score: 0.955 | F1 score: 0.957 |
| **DDcGAN-GSOM (Proposed)** | **Accuracy: 93.18%** | **Accuracy: 94.55%** | **Accuracy: 96.02%** | **Accuracy: 97.18%** | **Accuracy: 98.29%** |
| | **Precision: 92.12%** | **Precision: 91.59%** | **Precision: 93.45%** | **Precision: 96.98%** | **Precision: 95.80%** |
| | **F1 score: 0.933** | **F1 score: 0.928** | **F1 score: 0.947** | **F1 score: 0.982** | **F1 score: 0.975** |

**Table 6. Comparative analysis of DDcGAN-GSOM's Energy Consumption, Throughput, and Dealy.**

| Methods | Malicious Nodes (10%) | Malicious Nodes (20%) | Malicious Nodes (30%) | Malicious Nodes (40%) | Malicious Nodes (50%) |
|---|---|---|---|---|---|
| GRU-RNN | Energy consumption (joules): 97.63 | Energy consumption (joules): 85.57 | Energy consumption (joules): 75.84 | Energy consumption (joules): 62.94 | Energy consumption (joules): 54.4 |
| | Throughput (bytes): 6796.54 | Throughput (bytes): 15731.99 | Throughput (bytes): 22596.85 | Throughput (bytes): 32155.99 | Throughput (bytes): 42459.25 |
| | Delay(s): 584.0 | Delay(s): 513.25 | Delay(s): 442.5 | Delay(s): 371.75 | Delay(s): 301.0 |
| MLP | Energy consumption (joules): 94.21 | Energy consumption (joules): 81.12 | Energy consumption (joules): 73.72 | Energy consumption (joules): 64.33 | Energy consumption (joules): 52.81 |
| | Throughput (bytes): 6267.78 | Throughput (bytes): 16687.25 | Throughput (bytes): 25388.67 | Throughput (bytes): 33879.01 | Throughput (bytes): 42284.15 |
| | Delay(s): 550.67 | Delay(s): 479.92 | Delay(s): 409.17 | Delay(s): 338.42 | Delay(s): 267.67 |
| RNN-SDR | Energy consumption (joules): 91.62 | Energy consumption (joules): 81.21 | Energy consumption (joules): 70.47 | Energy consumption (joules): 62.62 | Energy consumption (joules): 51.5 |
| | Throughput (bytes): 8858.09 | Throughput (bytes): 16624.56 | Throughput (bytes): 27733.19 | Throughput (bytes): 36789.4 | Throughput (bytes): 45951.72 |
| | Delay(s): 517.33 | Delay(s): 446.58 | Delay(s): 375.83 | Delay(s): 305.08 | Delay(s): 271.75 |
| **DDcGAN-GSOM (Proposed)** | **Energy consumption (joules): 88.05** | **Energy consumption (joules): 81.19** | **Energy consumption (joules): 69.9** | **Energy consumption (joules): 57.32** | **Energy consumption (joules): 49.17** |
| | **Throughput (bytes): 11983.23** | **Throughput (bytes): 18046.45** | **Throughput (bytes): 29680.31** | **Throughput (bytes): 36065.05** | **Throughput (bytes): 45965.63** |
| | **Delay(s): 484.0** | **Delay(s): 413.25** | **Delay(s): 342.5** | **Delay(s): 271.75** | **Delay(s): 201.0** |

environments. It adapts to evolving threats by refining detection capabilities and ensuring long-term protection against advanced and unknown cyberattacks.

The findings as listed in Tables 5–7 provide a significant milestone, which gets further curated with the tabulation, i.e., the performance comparison of the proposed system with state-of-the-art approaches depicted in Table 8. The results for accuracy and precision of the proposal put forward in this research article, compared to contemporary algorithms like LSTM [21], MLP [25], RNN [27], CFCD [28], and H-ELM [30] which specialize in both accuracy and precision, show a striking resemblance. Thus, this finding firmly establishes the significance and importance of our proposal in the context of multilayered SDN security with MAC authentication.

## 5. Conclusion

This study introduces a MAC address-based authentication system for intrusion detection in Software-Defined Networks (SDNs), utilizing advanced methodologies to strengthen network security. By integrating Dual Discriminator Conditional Generative Adversarial Networks (DDcGAN) with Growing Self-Organizing Maps (GSOM), the system effectively detects and mitigates a range of attacks, including Distributed Denial of Service (DDoS) and scanning attacks. The DDcGAN technique generates synthetic network traffic that closely resembles real traffic, enhancing accuracy and reducing false positive rates. Compared to state-of-the-art methods such as GRU-RNN, MLP, and RNN-SDR, the proposed system exhibits superior performance in terms of accuracy, true positive rate, false alarm rate, and network throughput. A critical aspect of the system is its MAC address-based packet authentication, which classifies packets as legitimate or malicious, ensuring only legitimate packets are processed. Additionally, the use of the Four-Q Curve for user authentication addresses Man-in-the-Middle attacks, bolstering secure network access. The Sheep Flock Optimization Algorithm (SFOA) optimizes the DDcGAN model, further enhancing its efficiency in identifying intrusions. The system's implementation on

**Table 7. Comparative analysis of DDcGAN-GSOM's True Positive Rate, False Alarm Rate, and Specificity.**

| Methods | Malicious Nodes (10%) | Malicious Nodes (20%) | Malicious Nodes (30%) | Malicious Nodes (40%) | Malicious Nodes (50%) |
|---|---|---|---|---|---|
| GRU-RNN | True Positive Rate: 85.37% | True Positive Rate: 87.7% | True Positive Rate: 89.23% | True Positive Rate: 90.85% | True Positive Rate: 92.16% |
| | False alarm rate: 18.8% | False alarm rate:16.23% | False alarm rate: 13.6% | False alarm rate: 9.66% | False alarm rate: 8.46% |
| | Specificity: 64.0% | Specificity: 64.75% | Specificity: 72.2% | Specificity: 73.25% | Specificity: 79.0% |
| MLP | True Positive Rate: 87.41% | True Positive Rate: 89.06% | True Positive Rate: 91.62% | True Positive Rate: 93.1% | True Positive Rate: 94.96% |
| | False alarm rate: 16.6% | False alarm rate: 13.15% | False alarm rate: 11.97% | False alarm rate: 8.44% | False alarm rate: 5.18% |
| | Specificity: 66.0% | Specificity: 70.75% | Specificity: 78.5% | Specificity: 78.25% | Specificity: 82.0% |
| RNN-SDR | True Positive Rate: 89.52% | True Positive Rate: 92.22% | True Positive Rate: 93.83% | True Positive Rate: 94.96% | True Positive Rate: 96.68% |
| | False alarm rate: 4.62% | False alarm rate: 11.02% | False alarm rate: 8.02% | False alarm rate: 7.11% | False alarm rate: 3.05% |
| | Specificity: 78.0% | Specificity: 78.75% | Specificity: 81.5% | Specificity: 89.25% | Specificity: 93.0% |
| DDcGAN-GSOM (Proposed) | **True Positive Rate: 91.93%** | **True Positive Rate: 93.8%** | **True Positive Rate: 95.77%** | **True Positive Rate: 97.43%** | **True Positive Rate: 99.04%** |
| | **False alarm rate: 11.37%** | **False alarm rate: 8.59%** | **False alarm rate: 6.2%** | **False alarm rate: 5.48%** | **False alarm rate: 2.05%** |
| | **Specificity: 77.0%** | **Specificity: 83.75%** | **Specificity: 86.5%** | **Specificity: 87.25%** | **Specificity: 97.0%** |

**Table 8. Performance comparison of proposed system with state-of-the-art approaches.**

| Techniques | Accuracy (%) | Precision (%) |
|---|---|---|
| LSTM [21] | 97.1 | – |
| MLP [25] | 95.01 | 95.46 |
| GRU-RNN [26] | 89 | 99 |
| RNN [27] | 97.39 | 92 |
| CFCD [28] | 95.5 | 95.5 |
| H-ELM [30] | 84.29 | 94.18 |
| **Proposed (DDcGAN-GSOM)** | **98.29** | **95.8** |

the Ns-3 network simulator validates its effectiveness, demonstrating significant improvements over existing methods. The findings underline the potential of the proposed system to revolutionize intrusion detection in SDNs. The DDcGAN model's effectiveness relies on the quality and diversity of training data, which can be affected by the biased data. To improve its effectiveness, real-time network traffic data and active learning strategies can be implemented. Regularization techniques and monitoring can mitigate adversarial attacks. Integrating DDcGAN with other machine learning models like Graph Neural Networks can improve its ability to detect sophisticated network attacks in dynamic SDN environments. Federated learning techniques could enable distributed SDN devices to train models without sharing sensitive data. Future research should explore diverse datasets, privacy-preserving mechanisms, blockchain, and trust-based frameworks to enhance robustness, generalization, and data confidentiality in sensitive network environments.

## Supporting information

**S1 Table. Input and Selected features for data packet transmission.**
(DOCX)

**S1 Fig. Flowchart of the sheep flock optimization algorithm.**
(DOCX)

**S1 File. S1 Algorithm. GSOM algorithm for categorizing suspicious packets.**
(DOCX)

## Acknowledgments

The authors are grateful to the management of Vellore Institute of Technology (VIT), Vellore for providing all the facilities to carry out this research article. Due acknowledgement must be given to the Lab Incharge of Advanced Digital Signal Processing Lab (Technology Tower-231) for providing necessary space to conduct experiments.

## Author contributions

**Investigation:** Nanavath Kiran Singh Nayak.

**Methodology:** Nanavath Kiran Singh Nayak.

**Writing – original draft:** Nanavath Kiran Singh Nayak.

**Writing – review & editing:** Budhaditya Bhattacharyya.

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
