## [Decision Letter · Decision Letter 0]

7 Nov 2024

Dear Dr. Bhattacharyya,

Thank you for submitting your manuscript to PLOS ONE. After careful consideration, we feel that it has merit but does not fully meet PLOS ONE’s publication criteria as it currently stands. Therefore, we invite you to submit a revised version of the manuscript that addresses the points raised during the review process.

We look forward to receiving your revised manuscript.

Kind regards,

Muhammad Anwar, Ph.D.

Academic Editor

PLOS ONE

Journal Requirements:

4. We note that Figure 1 in your submission contain copyrighted images. All PLOS content is published under the Creative Commons Attribution License (CC BY 4.0), which means that the manuscript, images, and Supporting Information files will be freely available online, and any third party is permitted to access, download, copy, distribute, and use these materials in any way, even commercially, with proper attribution. For more information, see our copyright guidelines: http://journals.plos.org/plosone/s/licenses-and-copyright.

Reviewers' comments:

Reviewer's Responses to Questions

**Comments to the Author**

1. Is the manuscript technically sound, and do the data support the conclusions?

Reviewer #1: Yes

Reviewer #2: No

Reviewer #3: Partly

2. Has the statistical analysis been performed appropriately and rigorously?

Reviewer #1: Yes

Reviewer #2: No

Reviewer #3: Yes

3. Have the authors made all data underlying the findings in their manuscript fully available?

Reviewer #1: Yes

Reviewer #2: No

Reviewer #3: Yes

4. Is the manuscript presented in an intelligible fashion and written in standard English?

Reviewer #1: Yes

Reviewer #2: Yes

Reviewer #3: Yes

Reviewer #1: Here are my comments:

1. The current title, "An authentication system based on MAC addresses for multilayered SDN intrusion detection using dual discriminator conditional GAN and growing self-organizing maps," is quite detailed but could be more concise and specific. Here are some suggestions to make the title more focused and clearer:

a. "MAC Address Authentication for SDN Security: A Dual Discriminator GAN and Self-Organizing Maps Approach".

b. "Intrusion Detection in SDN Using MAC-Based Authentication and Conditional GANs"

c. "Multilayered SDN Security with MAC Authentication and GAN-Based Intrusion Detection"

2. Although the abstract is clearly organized, there are several unclear passages. Change the words from "Computer networks are susceptible to cyber-security intrusions" for "Computer networks are highly vulnerable to cyber-security intrusions." Additionally, as some readers may not be aware with the term "Four-Q curve authentication," quickly define it for them.

3. Further background on SDN and its significance in the modern networking environment has to be included in the introduction. Provide a succinct explanation of the drawbacks of conventional SDN designs as well as the importance of multilayer SDN in improving network security.

4. The literature study needs to provide a comparative examination of the many strategies that have been discussed. Talk about each method's benefits and drawbacks in further depth. Furthermore, it is important to include new research conducted beyond 2020 in order to maintain the currency of the literature review.

5. The methodology part needs to have an elaborate flowchart or diagram that demonstrates the architecture of the suggested system. This will make the procedure easier for readers to see and comprehend the processes needed.

6. Give a more thorough description of the relevance of the "Four-Q curve authentication" procedure in the suggested system.

7. The DC-GAN technique should be explained in greater detail. Provide explanations of the dual discriminator's operation and how it improves intrusion detection accuracy using mathematical formulas and algorithms.

8. Give a thorough explanation of the Sheep Flock Optimization Algorithm (SFOA). Explain its function in DC-GAN optimization and the rationale behind selecting this specific method over alternative optimization strategies.

9. The ramifications of the results should be covered in more detail in the discussion section. Talk about the practical applications of the suggested system and how it could affect SDN security.

10. Discuss any potential drawbacks or restrictions with the suggested system and offer ideas for workarounds or new directions for investigation.

11. The main conclusions and contributions of the study should be succinctly summarized in the conclusion. Stress the newness and importance of the suggested system for improving SDN security.

12. Provide targeted recommendations for future research trajectories to improve the suggested system even further and fill up any gaps.

Reviewer #2: 1. Authors should provide some brief numerical results at the end of the abstract to demonstrate that the proposed approach achieves SOTA performance.

2. The authors should provide the reference when DCGAN was first introduced, rather than waiting until line 247. In addition, reference [33] is commonly known as DDcGAN, so why do the authors refer to it as DCGAN?

3. Abbreviations (e.g. MAC, DDoS, MLP, IoT, and others) need to be defined before being mentioned for the first time.

4. Each symbol in Equation 2 should be introduced, and its corresponding value and meaning should be clearly explained.

5. The paper lacks figures, and the authors should add them to paper.

6. F1-score is a metric that assesses model performance on a scale from 0 to 1. For instance, better to express the F1-score as 0.975 rather than 97.5%.

7. This paper lacks tables of experimental results.

8. Authors should provide a detailed explanation of the proposed figures. For instance, Figures 14 and 15 represent the Discriminator Loss and Generator Loss, respectively. However, the current descriptions lack clarity and specificity. To make the figures more meaningful, the authors should descript relevant numerical values and offer a more comprehensive interpretation.

Reviewer #3: The paper entitled “An authentication system based on MAC addresses for multilayered SDN intrusion detection using dual discriminator conditional GAN and growing self-organizing maps.” presents a novel four-Q curve authentication system based on MAC addresses for multilayered SDN intrusion detection system utilizing deep learning techniques to identify and prevent attacks. I have the following comments that could improve the paper's structure:

1. The abstract is satisfactory; however, it necessitates some enhancements. Authors should initially enhance the abstract's composition. Subsequently, it should explicitly illustrate the outcomes of the proposed scheme, including numerical values and percentages.

2. Authors should give additional consideration to certain aspects of the manuscript, as well as to the numerous errors and formatting issues that are present in the current version.

3. The current manuscript contains several typos and formatting issues, along with several considerations the authors should take into account, such as:

-Page 2: " Therefore, by implementing an effective IDS in MAC-based SDN system, adversaries can be reduced or eliminated".

The authors should check for similar mistakes and revise them.

4. I advise the authors to revise the quality of all figures.

5. The proposed solution by the authors suggests the following: First, in the infrastructure layer, deep learning models get trained with more robustness. In addition, in the control plane layer, the local discriminator consists of a leaky ReLU activation function, two batch normalization layers, and five convolutional layers. The discriminator system includes four fully connected layers. Lastly, the application layer contains an unsupervised neural network of the GSOM type based on the SOM methodology. My question is, is it a practical solution to use all these complex techniques within each part? Please discuss in detail.

6. Authors are advised to compare their work with the literature in the result section.

**Do you want your identity to be public for this peer review?** For information about this choice, including consent withdrawal, please see our Privacy Policy

Reviewer #1: **Yes: ** Radhwan Mohamed Abdullah

Reviewer #2: No

Reviewer #3: No

---

## [Author Response · Author response to Decision Letter 1]

10 Dec 2024

We have provided the detailed responses to each comment and explain the revisions made in the manuscript in the file titled Response to the Reviewer Comments.

---

## [Decision Letter · Decision Letter 1]

13 Jan 2025

Dear Dr. Bhattacharyya,

Thank you for submitting your manuscript to PLOS ONE. After careful consideration, we feel that it has merit but does not fully meet PLOS ONE’s publication criteria as it currently stands. Therefore, we invite you to submit a revised version of the manuscript that addresses the points raised during the review process.

We look forward to receiving your revised manuscript.

Kind regards,

Muhammad Anwar, Ph.D.

Academic Editor

PLOS ONE

Reviewers' comments:

Reviewer's Responses to Questions

**Comments to the Author**

Reviewer #2: (No Response)

Reviewer #3: All comments have been addressed

2. Is the manuscript technically sound, and do the data support the conclusions?

Reviewer #2: Yes

Reviewer #3: Yes

3. Has the statistical analysis been performed appropriately and rigorously?

Reviewer #2: Yes

Reviewer #3: Yes

4. Have the authors made all data underlying the findings in their manuscript fully available?

Reviewer #2: No

Reviewer #3: Yes

5. Is the manuscript presented in an intelligible fashion and written in standard English?

Reviewer #2: No

Reviewer #3: Yes

Reviewer #2: 1. Why did the authors abbreviate Dual-Discriminator Conditional Generative Adversarial Network [36] as DC-GAN instead of DDcGAN?

2. Authors should revise their paper carefully to avoid writing errors: e.g., single and double inverted commas are used in a confusing way, why are there 'L_loss' and "L_loss"?

The multiplication sign should not be replaced by word "x".

In addition, the symbolic fonts of the equation need to be checked and revised again.

3. Better to highlight the best performances in bold, or in a different color (e.g. red) to make the tables easier to read.

Reviewer #3: The authors respond to every comment. Other than that, I have no concerns. The paper can be accepted as it is.

**Do you want your identity to be public for this peer review?** For information about this choice, including consent withdrawal, please see our Privacy Policy

Reviewer #2: No

Reviewer #3: No

---

## [Author Response · Author response to Decision Letter 2]

5 Feb 2025

Thank you for your insightful observation.

---

## [Decision Letter · Decision Letter 2]

17 Aug 2025

Multilayered SDN Security with MAC Authentication and GAN-Based Intrusion Detection

PONE-D-24-24864R2

Dear Dr. Bhattacharyya,

We’re pleased to inform you that your manuscript has been judged scientifically suitable for publication and will be formally accepted for publication once it meets all outstanding technical requirements.

Kind regards,

Muhammad Anwar, Ph.D.

Academic Editor

PLOS ONE

Additional Editor Comments (optional):

Reviewers' comments:

Reviewer's Responses to Questions

**Comments to the Author**

Reviewer #2: All comments have been addressed

2. Is the manuscript technically sound, and do the data support the conclusions?

Reviewer #2: Yes

3. Has the statistical analysis been performed appropriately and rigorously?

Reviewer #2: Yes

4. Have the authors made all data underlying the findings in their manuscript fully available?

Reviewer #2: (No Response)

5. Is the manuscript presented in an intelligible fashion and written in standard English?

Reviewer #2: Yes

Reviewer #2: (No Response)

**Do you want your identity to be public for this peer review?** For information about this choice, including consent withdrawal, please see our Privacy Policy

Reviewer #2: No

---

## [Editor Report · Acceptance letter]

PONE-D-24-24864R2

PLOS ONE

Dear Dr. Bhattacharyya,

I'm pleased to inform you that your manuscript has been deemed suitable for publication in PLOS ONE. Congratulations! Your manuscript is now being handed over to our production team.

Kind regards,

on behalf of

Dr. Muhammad Anwar

Academic Editor

PLOS ONE